computational chemistry

molecular dynamics, aluminium, peptide, semi-empirical

**Author for correspondence:**
James A. Platts
e-mail: platts@cardiff.ac.uk

This article has been edited by the Royal Society of Chemistry, including the commissioning, peer review process and editorial aspects up to the point of acceptance.

# Quantum chemical molecular dynamics and metadynamics simulation of aluminium binding to amyloid-β and related peptides

## James A. Platts

School of Chemistry, Cardiff University, Park Place, Cardiff CF10 3AT, UK

(iD) JAP, 0000-0002-1008-6595

We report semi-empirical tight-binding simulations of the interaction between Al(III) and biologically relevant peptides. The GFN2-XTB method is shown to accurately reproduce previously reported and density functional theory (DFT)-calculated geometries of model systems. Molecular dynamics simulations based on this method are able to sample peptide flexibility over timescales of up to nanoseconds, but these timescales are insufficient to explore potential changes in metal–peptide binding modes. To achieve this, metadynamics simulations using root mean square deviation as a collective variable were employed. With suitably chosen biasing potentials, these are able to efficiently explore diverse coordination modes, for instance, through Glu and/or Asp residues in a model peptide. Using these methods, we find that Al(III) binding to the N-terminal sequence of amyloid-β is highly fluxional, with all acidic sidechains and several backbone oxygens participating in coordination. We also show that such simulations could provide a means to predict *a priori* possible binding modes as a precursor to longer, atomistic simulations.

## 1. Introduction

Alzheimer's disease (AD) is one of the greatest healthcare challenges facing society. It is characterized by neuronal damage, associated with the formation of insoluble plaques and fibrils made up of amyloid-β (Aβ) peptides in the affected areas of the brain [1]. While aggregation of Aβ peptides is a key aspect of this process, the importance of metal ions in this process is increasingly recognized [2,3]. The primary focus of most research here has been on naturally occurring metals, notably Cu(II) [4–7],

Zn(II) [8–11] and Fe(II) [12,13], but studies over several decades have also implicated Al(III) in AD onset [14–17]. Aluminium is not one of the essential elements of human biology, but technological applications mean that human exposure to it has increased in the last century [18]. Exley and co-workers have shown that Al(III) has the ability to bind to Aβ and affect its conformational behaviour [14], promote aggregation [16] and help in the formation of reactive oxygen species [19]. Other studies report that Al(III) can be found in senile plaques from AD patients' brains [20] and that it forms smaller oligomers than naturally occurring metals [21].

Computer simulation has proved a valuable complement to experiment in exploring metal–Aβ interactions [22], but applications to Al(III) binding are scarce. A series of studies from Mujika et al. used density functional theory (DFT) to examine the pro-oxidant activity of Al [19], its binding to biomolecules [23] including metal-transport proteins [24,25] and neurotransmitters [26]. Most relevant for this work, the same group showed that Al(III) has a clear preference for anionic sidechains of peptides over backbone carbonyl or water [27], and that coordination numbers of 5 or 6 are preferred over lower values [28]. The latter study also sets out a template-based method for location of likely binding modes of Aβ, identifying Glu3, Asp7 and Glu11 as the preferred sites of interaction. Recently, our group used atomistic molecular dynamics (MD) to explore how Al(III) binding to Aβ in the identified coordination mode affects peptide structure and dynamics, finding that this ion promotes the formation of helical structures and disrupts the salt-bridge network [29]. However, even over the course several microseconds of MD trajectory, we found little or no evidence of disruption of the coordination sphere from that used at the outset, despite the use of a non-bonded model of ion binding that should, in principle, allow this to change. It was not clear from this study whether this behaviour is truly representative of Al-Aβ binding, or whether it stemmed from the chosen force-field model and/or simulation protocol.

In this work, we take a different approach and use semi-empirical tight-binding methods [30] to model Al(III) binding to Aβ and some related model peptides. We aim to ascertain whether such methods might represent a useful 'halfway-house' between DFT and atomistic modelling in modelling flexible peptides and their interactions with metal ions. Key to this is the fully self-consistent nature of the method, in which electronic structure of the metal–peptide complex is calculated at every step of a simulation, and thus able to respond to changes in environment and hence alter coordination modes. Alongside this, their speed means that dynamical simulation over relevant timescales becomes feasible, especially when coupled with metadynamics to move trajectories out of potential energy wells that could otherwise trap simulations, as demonstrated recently by Grimme [31]. In order to keep the calculations tractable, the full Aβ peptide was truncated to the N-terminal 1–16 residues that constitute the metal-binding region.

We have several reasons for choosing Al(III) for study: firstly, it may have direct biological relevance to AD and other disorders, as outlined above. Secondly, it is less studied than many other metal ions, so any new information on potential binding modes should be of interest to researchers in the field of metal–Aβ interactions. Thirdly, incorporation of di- and tri-valent ions into atomistic molecular mechanics simulations is not always straightforward, with limited transferability between different force fields, mixing rules and simulation conditions. One recent study proposed [32] different non-bonded parameters for prediction of hydration free energy or ion–oxygen distances of divalent cations, or a compromise between these. Others show that standard parameters for Ca(II) yield significant errors in some environments [33]. These issues are assigned to lack of charge transfer, polarization and potential covalent effects in standard non-bonded models. By comparing results obtained previously with such models with those from a self-consistent quantum mechanical approach, we aim to shed new light on suitable simulation protocols for such problems.

# 2. Computational methods

All tight-binding calculations were carried out using the GFN2-XTB method [34], running on a 28-core workstation equipped with Intel i9 processors and 32 Gb RAM. Geometry optimization used defaults for convergence, and for smaller systems confirmed as true minima by numerical harmonic frequency calculation. MD simulations were performed in the NVT ensemble at 310 K, using a Berendsen thermostat [35], with selected bonds restrained at their optimized lengths by means of the SHAKE algorithm [36]. Timesteps varied from 1 to 4 fs, depending on restrained bonds, the latter being facilitated by use of fictitious hydrogen mass of 4 amu. Metadynamics simulations used the same set-up, typically with 4 fs timestep, moving the simulation into new areas of phase space by the addition

**Table 1.** Comparison of DFT with GFN2-XTB optimized geometry of **Al-A** and **Al-AADAA** (Å).

| | A | | AADAA | |
|---|---|---|---|---|
| | DFT | GFN2-XTB | DFT | GFN2-XTB |
| Al–O$_{Asp}$ | 1.827 | 1.817 | 1.800 | 1.757 |
| Al–O$_{H2O}^a$ | 1.910 (0.025) | 1.875 (0.025) | 1.921 (0.019) | 1.886 (0.008) |
| AlOH…O | 1.497 | 1.606 | 1.627, 1.795 | 1.734, 1.737 |
| RMSD$^b$ | | 0.115 | | 2.495 |

$^a$Reported as mean (s.d.).
$^b$Root mean square deviation between DFT and GFN2-XTB structures.

of a biasing potential based on root mean square deviation (RMSD) as the collective variable, as described by Grimme [31]. DFT calculations employed the Gaussian09 package, using the B3LYP-D3(BJ) [37–39] functional and def2-TZVP basis set [40]. Analysis of trajectories used the VMD [41] and cpptraj [42] packages. All amino acids were constructed in appropriate protonation states for physiological pH, i.e. negative for Asp and Glu, positive for Arg and Lys and neutral for His. In principle, this could change over the course of simulations, but in practice, use of SHAKE restraints mean that this will not occur, and protonation states remain fixed throughout.

## 3. Results

It is important to assess the suitability of the GFN2-XTB method for the description of Al-peptide binding. To do so, we adopt two model systems previously used by Mujika *et al.* [27], namely Asp-Al(H$_2$O)$_5$ and Ala-Ala-Asp(Al(H$_2$O)$_5$)-Ala-Ala, denoted **Al-A** and **Al-AADAA**, respectively. Both were fully optimized at DFT and GFN2-XTB level, within implicit aqueous solvent. In both cases, the coordination bond to Asp is well described by the semi-empirical method, at 0.01–0.03 Å shorter than DFT. Bonds to water show slightly greater deviation, but are still within 0.05 Å of the DFT benchmark. Both complexes exhibit hydrogen bonding: **Al-A** contains a single H-bond between water and non-coordinated carboxylate oxygen, predicted to be within 0.1 Å of DFT by GFN2-XTB. **Al-AADAA** forms two hydrogen bonds between water and backbone carbonyl, which are again *ca* 0.1 Å longer from GFN2-XTB. RMSD is small for **Al-A**, indicating excellent overall agreement between methods, but rather larger for **Al-AADAA** at almost 2.5 Å. Closer analysis indicates that most of this stems from the peptide: restricting the calculation to Al, carbonyl O plus water yields RMSD = 0.932 Å, demonstrating the suitability of GFN2-XTB for the description of peptide–Al(III) complexes (table 1).

The speed and accuracy of GFN2-XTB lend it to dynamical simulation, so several 1 ns MD simulations were performed on **Al-AADAA**, starting from the optimized geometry. The first of these used timesteps of 2 fs, along with SHAKE restraints on bonds to hydrogen, in the NVT ensemble using the Berendsen thermostat to maintain temperature of 310 K. After a period of equilibration of approximately 250 ps, temperature and total energy were stable over the simulation (see electronic supplementary material, figure S1). To enhance sampling efficiency, a further simulation restrained all bonds except those to Al with a fictitious hydrogen mass of 4.0 amu, allowing a timestep of 4 fs. Equilibration is in this case observed in *ca* 200 ps, after which temperature and total energy were stable over the remainder of the simulation. Greater changes were observed in potential energy, which dropped sharply over the first 100 ps of simulation before stabilizing; RMSD increased over a similar timescale, before reaching a plateau around 2.5 Å (electronic supplementary material, figure S2). These changes are associated with changes in the peptide backbone to accommodate more hydrogen bonds between coordinated water and remote backbone carbonyls, along with movement away from the extended starting geometry towards the preferred bent geometry of the peptide. This is also evident in the end-to-end distance, which falls rapidly from an initial value of 21.6 Å, reaching less than 5 Å within 100 ps, then oscillating around *ca* 9 Å for the rest of the trajectory.

Although conventional MD successfully samples peptide conformation, even for these small peptides nanosecond simulations require substantial computing resources while leaving the Al coordination sphere largely unchanged. We therefore turn to metadynamics in order to enhance sampling and move the complex into new conformations and/or coordination modes. Following Grimme, we use

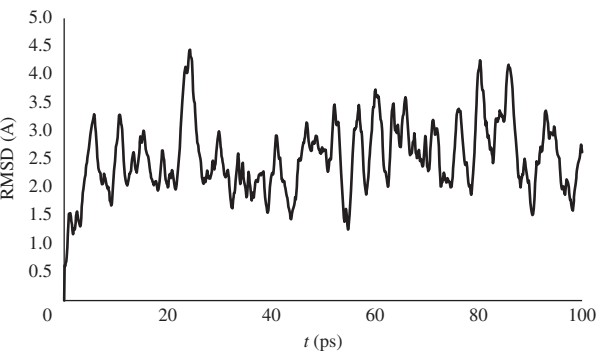

**Figure 1.** RMSD from 100 ps metadynamics simulation of **Al-AADAA**.

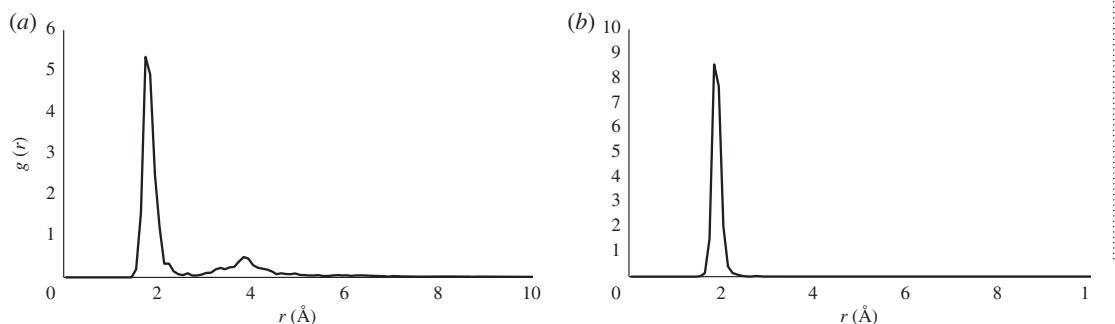

**Figure 2.** RDF for (*a*) Al–O$_{pept}$ and (*b*) Al–O$_{wat}$ distances in **Al-EAAAD**.

RMSD from the starting structure as the collective variable, enclosing the system within a wall potential to prevent decomposition. A 100 ps of metadynamics was carried out on **Al-AADAA**, adding Gaussian potentials every picosecond and a pushing constant $k_i/N = 0.001\ E_h$, with 4 fs timestep, SHAKE on all non-metal bonds and H mass = 4 amu. Such a simulation explored a similar set of conformations as conventional MD but in a much reduced time frame, with the end-to-end distance reaching *ca* 5 Å within 50 ps. RMSD from initial structure rises quickly before exhibiting oscillatory behaviour after approximately 10 ps, repeatedly reaching larger values than were observed over a 10 times longer conventional MD simulation (figure 1). However, the Al(III) coordination mode remains intact throughout, indicating that the bias potential in this case is applied mainly to the peptide.

As a more challenging example, we examined **Al-EAAAD**, in which Al(III)(H$_2$O)$_5$ was initially coordinated to OE2 of the N-terminal glutamic acid. Conventional MD showed similar behaviour to that described above, reaching equilibration in *ca* 100 ps before settling into a stable conformation for the remainder of the simulation. Metadynamics with the same bias potential as used for **Al-AADAA** behaved similarly, exploring the conformational freedom of the peptide more than conventional MD but keeping Al-coordination intact. However, an increased pushing constant of $k_i/N = 0.025\ E_h$ gave rise to quite different behaviour. After a brief period of equilibration in which only the peptide moved appreciably, a water molecule was displaced from Al(III) coordination by the backbone O of Glu1 after *ca* 10 ps. The resulting chelated Glu1-Al(H$_2$O)$_4$ structure persisted for around 50 ps, before further waters were displaced, first by OE1 of Glu1, then by backbone O of alanines, and eventually carboxylate of Asp5. By the end of a 70 ps metadynamics run, no water was bound directly to Al(III), with coordination supplied by sidechains of Glu1 (monodentate) and Asp5 (bidentate) as well as backbone O of Glu1 and Ala3. During the entire simulation, every peptide O atom spends at least 5ps within 2 Å of the metal. The resulting radial Al–O distribution functions (RDFs) for peptide and water oxygens are shown in figure 2: both show notable peaks around 2 Å, slightly lower for peptide than for water, while the former also shows a broad peak around 4 Å, to outer-shell hydrogen-bonded contacts mediated through inner-shell waters. The integration of RDF values indicate that on average, peptide O coordination number is *ca* 2.4 while water coordination number is *ca* 3.2, yielding an overall value of slightly less than 6. This compares well with Mujika *et al*'s findings that penta- and hexa-coordinated Al(III) ions are preferred, and that carboxylate ligands are particularly favourable binders.

Encouraged by this performance of RMSD-biased metadynamics for smaller peptides, we turned our attention to the interaction of Al(III) to the N-terminal sequence of amyloid-β, denoted **Al-Aβ16**. To do

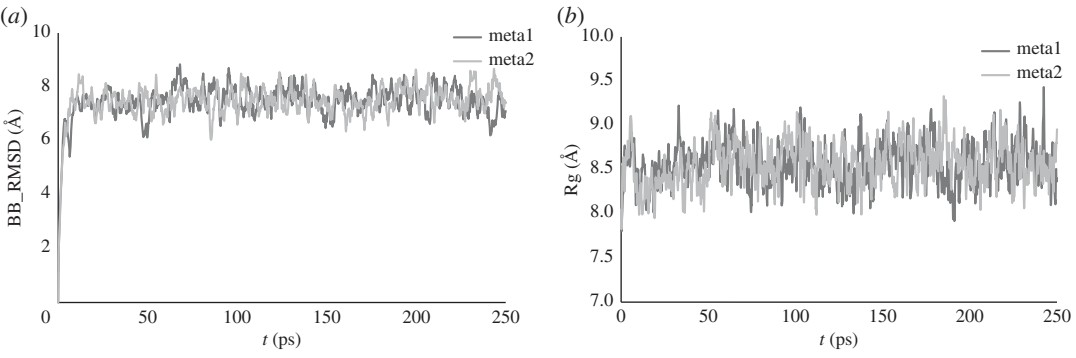

**Figure 3.** Backbone RMSD (*a*) and Rg (*b*) over two separate 250 ps metadynamics simulations on **Al-AB16**.

so, we took as a starting point the structure of Al-Aβ28 in explicit aqueous solvent reported by Mujika *et al.* [28], truncating this to the N-terminal 16 amino acids along with Al, bound initially through backbone (1.90 Å) and sidechain (monodentate 1.73 Å) of Glu3, sidechain of Asp7 (monodentate 1.66 Å), and backbone (1.91 Å) and sidechain (bidentate 1.73 and 1.75 Å) of Glu11, in approximately octahedral geometry. Twenty water molecules within 10 Å of the Al(III) ion were also retained. GFN2-XTB optimization of this system retained the coordination sphere of Al and the peptide conformation, with RMSD from starting point of 1.40 Å. A 1 ns conventional MD simulation with settings as outlined above (4 fs timestep, SHAKE on all non-metal bonds) proceeded smoothly, with backbone RMSD rising smoothly to 4.5 Å within 50 ps and remaining approximately constant thereafter, with no perturbation of Al coordination (electronic supplementary material, figure S5).

Two separate metadynamics simulations, each of 250 ps, starting from the same optimized geometry but with different initial velocities, were carried out, with pushing constant $k_i/N = 0.006$ $E_h$. Backbone RMSD rose rapidly at first, before stabilizing to values oscillating around 7.5 Å after *ca* 20 ps, while Rg stabilized to an average value of 8.6 Å after the same time (figure 3). Behaviour was broadly similar to that noted for **Al-EAAAD**: an initial period of around 50 ps involved motion of peptide and water, with Al binding remaining stable. After this point, Al–O bonds dissociated and others enter the coordination sphere, leading to coordination numbers as low as 4 and as high as 7, with the mean value of 5.2 (electronic supplementary material, figure S7). On average, coordination from sidechain oxygens is twice as prevalent as that from backbone atoms (mean 3.4 versus 1.8), but as many as four backbone oxygens coordinate Al(III) in some frames. As well as providing evidence of the utility of metadynamics in simulating such systems, the broad similarity of the data from two independent simulations lends confidence that results are not strongly dependent on starting conditions.

Interrogation of Al–O distances reveals highly fluxional coordination: every sidechain oxygen, with the exception of Gln15, spent some time in contact with Al (defined as less than 2.75 Å, vide infra). This contact was fleeting for Asp1 (1% of frames), but persistent for Glu3, Asp7 and Glu11, without any being present for the whole of the simulations. That the sum of contact for these residues exceeds 100% indicates bidentate coordination is present: close contacts for both sidechain O atoms were found for 14, 17 and 27% of frames for Glu3, Asp7 and Glu11, respectively. Backbone oxygens also featured in Al coordination: the initial coordination through Glu3 persisted through much of the simulations, but in contrast that through Glu11 was rapidly (less than 10 ps) lost and returned only fleetingly. Backbone oxygens of Phe4 and Arg5 also made significant contact with Al, whereas those later in the sequence, and especially Val12 and beyond, were much less involved in coordination (table 2).

Combining Al–O distances into a radial distribution function, *g(r)*, sheds further light on Al(III) binding. Separate *g(r)* data for oxygens in the peptide and in water from three independent metadynamics runs are plotted in figure 4 (plots from individual simulations are reported in electronic supplementary material, showing essentially no difference for peptide contacts, and only minor ones for water). The former exhibits a sharp peak centred on 1.75 Å, extending to 2.75 Å, and a lower, shallower peak around 3.85 Å. The former is taken as the first coordination shell, which integrates to 5.4, an estimate of the average coordination number of the ion, and one that is in accord with DFT estimates of the preferred coordination numbers for Al(III) as being 5 and 6. The peak around 4 Å is associated with the non-bonded O of monodentate acidic sidechain ligands. A small peak is also found around 6 Å, which may be due to outer-shell coordination. *g(r)* for water also shows a peak around 2 Å, but the values are markedly lower, such that the contribution to

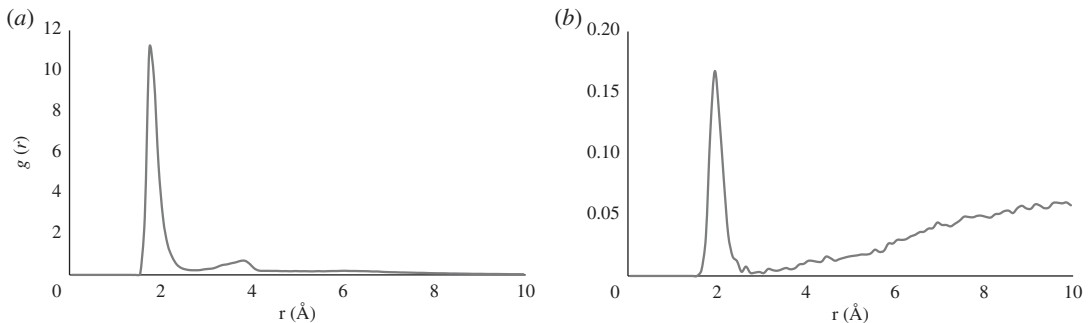

**Figure 4.** Radial distribution functions for Al−O distance from (*a*) peptide and (*b*) solvent.

**Table 2.** Percentage of simulation frames contributing to the first coordination sphere.

|        | sidechain[a] | backbone |
|--------|--------------|----------|
| Asp1   | 1/0          | 6        |
| Ala2   |              | 7        |
| Glu3   | 21/92        | 67       |
| Phe4   |              | 37       |
| Arg5   |              | 24       |
| His6   |              | 13       |
| Asp7   | 34/75        | 5        |
| Ser8   |              | 1        |
| Gly9   |              | 4        |
| Tyr10  | 0.0          | 8        |
| Glu11  | 74/53        | 6        |
| Val12  |              | 0        |
| His13  |              | 2        |
| His14  |              | 1        |
| Gln15  | 0            | 0        |
| Lys16  |              | 0        |

[a]Reported for $O\delta_1/O\delta_1$ or $O\varepsilon_1/O\varepsilon_2$ where appropriate.

coordination from solvent is just 0.07, i.e. effectively ruling out solvent participation in coordination, at least within the timescale of these simulations.

A more challenging problem is to determine whether metadynamics simulations such as these can locate potential binding modes without being placed in suitable starting geometry. A model system consisting of $Al(H_2O)_6$ placed in the proximity of EAAAD, with deprotonated Glu and Asp, was constructed. Geometry optimization and conventional MD formed hydrogen bonds between water O–H and peptide oxygens, but did not perturb ion coordination. By contrast, metadynamics with a pushing constant $k_i/N = 0.025$ rapidly sampled alternative coordination modes. Within 15 ps, one water molecule was displaced by $O\delta$ of Asp, leading to a relatively stable monodentate adduct that persisted for 15 ps, at which point a second water molecule was displaced by the backbone O of Ala. Further waters were displaced from Al after around 50 ps of simulation, until by 65 ps, all were detached completely, after which point the simulation proceeded in similar fashion to that described above.

A similar procedure for **Al-AB16** was tested: $Al(H_2O)_6$ placed in the proximity of Aβ16, and optimized to form hydrogen bond contacts with N-terminal residues. Three metadynamics simulations, each of 250 ps in length and with identical settings except their initial velocities, were started from the optimized geometry. In all cases, backbone RMSD increased rapidly in the first 10 ps, before reaching a plateau of around 7 Å, due to motions corresponding to peptide flexibility and intermolecular contacts. For each simulation, the first *ca* 50 ps consisted of intact $Al(H_2O)_6$ bound

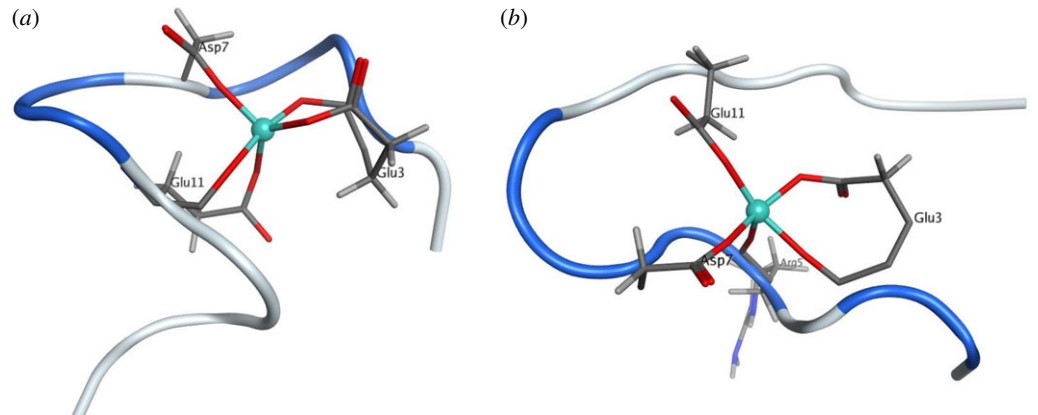

**Figure 5.** Endpoint of 1 ns conventional MD (*a*) and of 250 ps metadynamics (*b*) of **Al-AB16**. Al(III) is shown as a teal sphere, coordinating residues as stick models, and peptide backbone as ribbon (grey, coil; blue, turn).

through hydrogen bonds to various points on the peptide. After that, Al–O bonds to water ruptured: in runs 1 and 3, these were quickly replaced by peptide oxygens, but in run 2 low-valent $Al(H_2O)_n$ ($n = 1, 2, 3$) species persisted for *ca* 40 ps before bonds to peptide formed. Despite this, 100 ps is sufficient for the initial hydration sphere to be completely lost and Al(III) to be completely bound by the peptide, after which point behaviour is similar, both within these runs and to the simulations starting from bound Al reported above. In particular, all acidic sidechains spend some time coordinated to Al, as do several backbone oxygens in the N-terminal region. We also note that the starting point for these simulations appears to be irrelevant, as the first 50 ps effectively equilibrates the location of $Al(H_2O)_6$. This is evident in the distance between Al and N-terminal Cα, which rises from 6 Å in the optimized geometry to 16 Å after 20 ps, falls to 5 Å and rises again to 15 Å within 50 ps, before stabilizing to around 4 Å once Al is bound through acidic sidechains.

## 4. Discussion

The data reported show that GFN2-XTB is a useful method for the description of interactions of Al(III) with peptides, comparing well with DFT geometry for model peptides for a small fraction of the computational time required. This speed makes the dynamic simulation of such systems feasible on desktop computing resources. Such simulations proceed smoothly, while the use of restraints on bond lengths along with fictitious hydrogen mass allows timestep of 4 fs to further enhance efficiency. Conventional MD allows peptide conformation to change, but leaves coordination of the metal ion intact, even over multiple nanosecond simulation times. It is only when a biasing potential is introduced through metadynamics that we observe sampling of alternative coordination modes. These simulations typically undergo a short (20–50 ps) period of conformational flexibility of the peptide, before bonds to Al are ruptured and different atoms enter the inner coordination sphere. Ion binding is solely through oxygen, often of acidic sidechains but also of peptide backbone; however, where explicit water is present in simulations, it does not contribute significantly to ion binding. The ability of metadynamics to alter coordination mode is demonstrated in figure 5, which shows the endpoint of 1 ns conventional MD and of 250 ps metadynamics of **Al-AB16**. The former is little changed from the starting point, i.e. Al bound through the backbone and sidechain of Glu3 and Glu11, and sidechain only of Asp7, whereas metadynamics substantially alters the coordination sphere, ending with backbone O of Arg5 bound to Al, along with only sidechain atoms of the above-mentioned residues. In both cases, the peptide is largely unstructured, with short sequences between metal-binding residues identified as having turn character (blue in figure 5).

Simulations suggest that Al(III) coordination to Aβ16 is fluxional, the ion being capable of coordinating through a wide variety of O donors in the peptide. The question then arises, why was such behaviour not observed in microsecond length MD simulations using conventional, atomistic description of the peptide and ion? One can envisage that this stems from shortcomings in the non-bonded model of ion coordination employed, or from the simulation protocol employed, or both. As noted above, a non-bonded atomistic force field neglects details of charge transfer and polarization due to ion–peptide interactions. The metal centre parameter builder approach used in that work allows atomic charges on the peptide to include ion proximity effects, but not to adapt over the course of a simulation, such that a particular binding

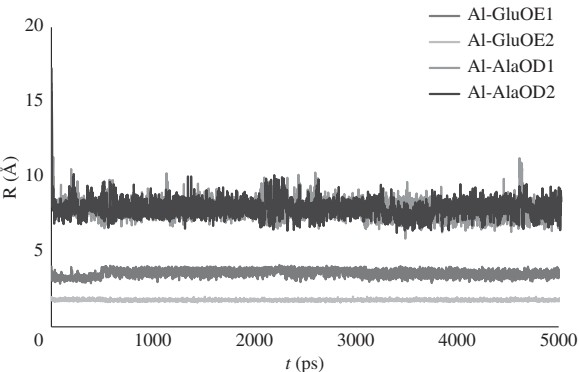

**Figure 6.** Al–O distances in **Al-EAAAD** over 5 ns conventional MD simulation.

mode may be 'locked in' by this procedure. This is apparent in the atomic charges obtained in our previous study, based on restrained electrostatic potential (RESP) analysis of B3LYP/6-31G(d) electron density. Within a monodentate carboxylate ligand, the O bound to Al was assigned much more negative charge (−1.02/−0.90 e) than the unbound one (−0.54/−0.55 e), whereas charges for bidentate carboxylate are more similar (−0.91 and −0.72 e). Similarly, a bound backbone O was more negative (−0.79 e) than its non-bound counterparts (−0.40 to −0.58 e). Such differences could prevent exploration of alternative coordination modes in atomistic simulation, especially since electrostatic forces are expected to dominate the interaction between these hard donor/acceptors.

However, our data also suggest another possible reason for lack of sampling: we find that, even with the ability for the method to adapt electronic structure to the ion's immediate environment, conventional MD does not alter the coordination mode established at the outset of the simulation. Of course, the simulations reported here are much shorter than those possible with purely atomistic methods, due to the greater computational overhead of the method. It is not feasible with current resources for us to determine whether microsecond or longer trajectories might eventually sample binding through other oxygens. We did, however, extend one conventional MD simulation on **Al-EAAAAD** to 5 ns (with all non-metal bonds restrained, 4 fs timestep). This was at the limit of the computing resources available to us. During this time, coordination of Al(III) exclusively through OE2 of Glu is retained: we find no evidence even of bidentate coordination, still less of Asp becoming involved in ion binding (figure 6). We conclude, therefore, that changes in coordination of Al(III) are 'rare events' on these timescales, such that enhanced sampling methods are required to move trajectories into new configurations.

It is appropriate at this stage to discuss possible limitations of this work. Firstly, despite the speed of the GFN2-XTB method, computational resources are a key consideration: it was necessary to truncate the Aβ sequence to the N-terminal 16 residues, and even with this short peptide, we were only able to generate a few nanoseconds of MD or metadynamics trajectories. This is insufficient to sample the whole conformational space of such a flexible peptide, which can take multiple microsecond or more timescales. Still further out of reach, at present at least, would be the simulation of larger peptides such as full-length amyloid-β and its oligomers that are implicated in the onset of Alzheimer's. We therefore see this approach as complementary to atomistic models, or the template approach taken by Mujika *et al*. In particular, the use of GFN2-XTB with metadynamics could be used to identify possible ion-binding modes in cases where little or no experimental data are available to guide the choice of residues bound to the metal ion. Short metadynamics runs could identify possible binding modes, from which longer atomistic simulations could be started.

Secondly, we have chosen Al(III) for study here due to its biological relevance, as well as potential challenges in modelling such ions within classical force-field approaches. Table 1 indicates that GFN2-XTB works well for Al-peptide complexes, but we have not yet tested this for other metal ions. In particular, transition metals such as copper, iron and zinc are known to be important in Alzheimer's and other neurodegenerative diseases. These display subtle d-orbital effects that are not relevant for Al(III): these are included in GFN2-XTB parameterization, but we have not provided any evidence for the balanced description of different metals' binding to Aβ. In addition, we have not addressed whether such simulations can be used to calculate binding (free) energy of the metal ion, and if any specificity for Al(III) over other ions can be demonstrated.

Thirdly, one issue that did not arise in this work but may be relevant in other cases is that of protonation state: all Asp and Glu residues were modelled as being deprotonated, ready to interact

with the 'hard' Al(III) ion. In principle, a method such as GFN2-XTB could model changes in protonation state, but distance restraints on bonds to hydrogen are incompatible with this. Moreover, using RMSD of all atoms as the collective variable in metadynamics may not be sufficient to allow deprotonation *and* metal binding to occur within simulations of the timescale reported here.

Fourthly, while RMSD has proved useful as the collective variable in this work, we have not tested alternatives to this. In atomistic modelling of flexible peptides such as Aβ, variables such as radius of gyration, backbone dihedral angles, helical and strand content, and coordination of one or more residues have been used in this role, and may provide more efficient, or at least different, sampling of coordination and conformation modes. Moreover, in this work, the collective variable was applied to all atoms, whereas our main interest is in the behaviour of Al and coordinating residues, such that restricting the biasing potential to a subset of atoms may be beneficial. We hope to report the results of such studies in future submissions.

# 5. Conclusion

We have demonstrated the utility of the semi-empirical tight-binding method GFN2-XTB for modelling the interactions of Al(III) with various peptides. Optimized geometries compare well with DFT benchmarks, and the speed of the method allows molecular dynamic simulations. Conventional MD allows changes in peptide conformation but leaves Al coordination intact, whereas metadynamics with a suitably chosen biasing potential is able to sample different coordination modes. Application of this approach to the N-terminal fragment of amyloid-β suggests that Al(III) binding is fluxional, with the majority of oxygens found in contact with the ion at some point during multiple 250 ps metadynamics simulations, leading to an estimated average coordination number of 5.2, with little or no water contributing to this value. We also show that suitable simulations are able to identify potential binding modes in unbiased fashion, starting from hydrogen-bonded contact between $Al(H_2O)_6$ and an extended peptide. We envisage this approach complementing atomistic, force-field bases MD, perhaps by identifying diverse starting points for longer timescale dynamics that can fully explore the conformational flexibility of peptides such as Aβ.

Data accessibility. Trajectories for MD and metadynamics simulations have been deposited on Zenodo with doi:10.5281/zenodo.3264898.

Competing interests. The author declares no competing interests.

Funding. This work was funded by the UK Engineering and Physical Sciences Research Council (EPSRC) under grant ref. no. EP/N016858/1.

Acknowledgements. The author is grateful to Dr Matthew Turner for careful proof-reading of the manuscript, and to Advanced Research Computing at Cardiff (ARCCA) for provision of computing resources.

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
