## [Reviewer comments · Royal Society Open Science]

Review History

RSOS-191143.R0 (Original submission)

Review form: Reviewer 1

Is the manuscript scientifically sound in its present form?

No

Are the interpretations and conclusions justified by the results?

Yes

Is the language acceptable?

Yes

Do you have any ethical concerns with this paper?

No

Have you any concerns about statistical analyses in this paper?

No

Recommendation?

Major revision is needed (please make suggestions in comments)

Comments to the Author(s)

James Platts as the only author, presents updated methods of metadynamics simulations using RMSD as a collective variable, to study the interaction of Al(III) and peptides. All the data is focused on the short peptides, including AI-EAAAD, AI-AB16. Full length beta-amyloid has its complicated 3D structure, has the strong interaction with other beta-amyloid monomer forming salt bridge as well as other metal ions. The specificity is not considered and discussed well. The buffer environment and pH transition is not included as well. Most important, the difference between beta-amyloid monomers and oligomers are not covered. The author also need to provide the evidence that the improved computer simulation can better interpret the interaction between beta-amyloid and metal, by solid bench experimental data.

Review form: Reviewer 2

Is the manuscript scientifically sound in its present form?

No

Are the interpretations and conclusions justified by the results?

No

Is the language acceptable?

Yes

Do you have any ethical concerns with this paper?

No

Have you any concerns about statistical analyses in this paper?

No

Recommendation?

Major revision is needed (please make suggestions in comments)

Comments to the Author(s)

The authors use a semi-empirical method GFN2-XTB for modelling the interactions of Al(III) with different peptides including Alzheimer's Abeta N-terminal fragment. The application of the employed method in studying metal and peptide interaction is an important and timely area of research. However, the authors fail to provide a comprehensive analysis and important insights from the simulation results. More analysis beyond RMSD is needed. Besides, they provide no validation of the used models, no discussion on the limitations of their approach, omit many important details, and provide no justice to the vast literature on this topic. Overall, I consider this study as weak, and recommend for a major revision.

1. The figures are designed very poorly. For instance, most of the figures have a distance axis denoted as "A" which should be Å.
2. Figure 3 labeling a and b are missing.
3. The irregular radial distribution function calculated for Al–O distance from solvent (Figure 3b) could be due to insufficient sampling. Replication of each experiment (at least 3 times) is needed to verify such irregular changes.
4. Figure 4 images taken from vmd snapshot need labeling. Label amino acids and increase the image resolution.

5. Authors compared conformational changes in peptide from their conventional MD and metadynamics results at different time scale. The images shown in figure 4 shows an unstructured peptide in both. Provide a superimpose structure derived from a fixed time scale of simulation.

Decision letter (RSOS-191143.R0)

05-Aug-2019

Dear Dr Platts:

Manuscript ID: RSOS-191143

Title: "Quantum Chemical Molecular Dynamics and Metadynamics Simulation of Aluminium Binding to Amyloid- β and Related Peptides"

Thank you for submitting the above manuscript to Royal Society Open Science. Your paper was sent to reviewers and their comments are included at the bottom of this letter.

In view of the concerns raised by the reviewers, the manuscript has been rejected in its current form. However, a new manuscript may be submitted which takes into consideration these comments.

Please note that resubmitting your manuscript does not guarantee eventual acceptance, and that your resubmission will be subject to peer review before a decision is made.

Your resubmitted manuscript should be submitted by 02-Feb-2020. If you are unable to submit by this date please contact the Editorial Office.

Yours sincerely,
Dr Ellis Wilde
Publishing Editor, Journals

On behalf of the Subject Editor Professor Anthony Stace and the Associate Editor Mr Andrew Dunn

REVIEWER REPORTS:

Associate Editor Comments to Author ():

RSC Associate Editor

Comments to the Author:

The reviewers indicate that extensive major revisions are required, and I believe that R&R would be most appropriate in this case, as significant experimental work is required which may not be possible in the major revisions timescale.

RSC Subject Editor

Comments to the Author:

(There are no comments.)

Reviewers' Comments to Author:

Reviewer: 1

Comments to the Author(s)

James Platts as the only author, presents updated methods of metadynamics simulations using RMSD as a collective variable, to study the interaction of Al(III) and peptides. All the data is focused on the short peptides, including AI-EAAAD, AI-AB16. Full length beta-amyloid has its complicated 3D structure, has the strong interaction with other beta-amyloid monomer forming salt bridge as well as other metal ions. The specificity is not considered and discussed well. The buffer environment and pH transition is not included as well. Most important, the difference between beta-amyloid monomers and oligomers are not covered. The author also need to provide the evidence that the improved computer simulation can better interpret the interaction between beta-amyloid and metal, by solid bench experimental data.

Reviewer: 2

Comments to the Author(s)

The authors use a semi-empirical method GFN2-XTB for modelling the interactions of Al(III) with different peptides including Alzheimer's Abeta N-terminal fragment. The application of the employed method in studying metal and peptide interaction is an important and timely area of research. However, the authors fail to provide a comprehensive analysis and important insights from the simulation results. More analysis beyond RMSD is needed. Besides, they provide no validation of the used models, no discussion on the limitations of their approach, omit many important details, and provide no justice to the vast literature on this topic. Overall, I consider this study as weak, and recommend for a major revision.

1. The figures are designed very poorly. For instance, most of the figures have a distance axis denoted as "A" which should be Å.
2. Figure 3 labeling a and b are missing.
3. The irregular radial distribution function calculated for Al–O distance from solvent (Figure 3b) could be due to insufficient sampling. Replication of each experiment (at least 3 times) is needed to verify such irregular changes.
4. Figure 4 images taken from vmd snapshot need labeling. Label amino acids and increase the image resolution.
5. Authors compared conformational changes in peptide from their conventional MD and metadynamics results at different time scale. The images shown in figure 4 shows an unstructured peptide in both. Provide a superimpose structure derived from a fixed time scale of simulation.

Author's Response to Decision Letter for (RSOS-191143.R0)

See Appendix A.

RSOS-191562.R0

Review form: Reviewer 1

Is the manuscript scientifically sound in its present form?

Yes

Are the interpretations and conclusions justified by the results?

Yes

Is the language acceptable?

Yes

Do you have any ethical concerns with this paper?

No

Have you any concerns about statistical analyses in this paper?

No

Recommendation?

Accept with minor revision (please list in comments)

Comments to the Author(s)

This study investigated the interactions of Al(III) with various peptides of A β using semi-empirical tight binding method GFN2-XTB. Binding free energies are widely used to identify the key residues of peptides interacting with other molecules, including metal ions. So binding free energy analysis and residue contributions of peptides to the interaction with Al(III) can be calculated and discussed. It is better of the author to discuss why the A β 1-16 was selected for MD, but not the hydrophobic core of A β 17-22?

Decision letter (RSOS-191562.R0)

28-Oct-2019

Dear Dr Platts:

Title: Quantum Chemical Molecular Dynamics and Metadynamics Simulation of Aluminium Binding to Amyloid- β and Related Peptides
Manuscript ID: RSOS-191562

Thank you for submitting the above manuscript to Royal Society Open Science. On behalf of the Editors and the Royal Society of Chemistry, I am pleased to inform you that your manuscript will

be accepted for publication in Royal Society Open Science subject to minor revision in accordance with the referee suggestions. Please find the reviewers' comments at the end of this email.

The reviewers and handling editors have recommended publication, but also suggest some minor revisions to your manuscript. Therefore, I invite you to respond to the comments and revise your manuscript.

Because the schedule for publication is very tight, it is a condition of publication that you submit the revised version of your manuscript before 06-Nov-2019. Please note that the revision deadline will expire at 00.00am on this date. If you do not think you will be able to meet this date please let me know immediately.

Best wishes,
Dr Laura Smith
Publishing Editor, Journals

RSC Associate Editor
Comments to the Author:
(There are no comments.)

Reviewer comments to Author:
Reviewer: 1

Comments to the Author(s)
This study investigated the interactions of Al(III) with various peptides of A β using semi-empirical tight binding method GFN2-XTB. Binding free energies are widely used to identify the key residues of peptides interacting with other molecules, including metal ions. So binding free energy analysis and residue contributions of peptides to the interaction with Al(III) can be calculated and discussed. It is better of the author to discuss why the A β 1-16 was selected for MD, but not the hydrophobic core of A β 17-22?

Author's Response to Decision Letter for (RSOS-191562.R0)

See Appendix B.

RSOS-191562.R1 (Revision)

Review form: Reviewer 1

Is the manuscript scientifically sound in its present form?

Yes

Are the interpretations and conclusions justified by the results?

Yes

Is the language acceptable?

Yes

Do you have any ethical concerns with this paper?

No

Have you any concerns about statistical analyses in this paper?

No

Recommendation?

Accept with minor revision (please list in comments)

Comments to the Author(s)

I recommend the publication of this manuscript after the author address the following points.

There are several minor defects in this manuscript, please check it carefully and correct them. For example, the repeat spelling of "with" in the first line on page 14.

Decision letter (RSOS-191562.R1)

02-Dec-2019

Dear Dr Platts:

Title: Quantum Chemical Molecular Dynamics and Metadynamics Simulation of Aluminium Binding to Amyloid- β and Related Peptides
Manuscript ID: RSOS-191562.R1

Thank you for submitting the above manuscript to Royal Society Open Science. On behalf of the Editors and the Royal Society of Chemistry, I am pleased to inform you that your manuscript will be accepted for publication in Royal Society Open Science subject to minor revision in accordance with the referee suggestions. Please find the reviewers' comments at the end of this email.

The reviewers and handling editors have recommended publication, but also suggest some minor revisions to your manuscript. Therefore, I invite you to respond to the comments and revise your manuscript.

Because the schedule for publication is very tight, it is a condition of publication that you submit the revised version of your manuscript before 11-Dec-2019. Please note that the revision deadline will expire at 00.00am on this date. If you do not think you will be able to meet this date please let me know immediately.

- 1) A text file of the manuscript (tex, txt, rtf, docx or doc), references, tables (including captions) and figure captions. Do not upload a PDF as your "Main Document".
- 2) A separate electronic file of each figure (EPS or print-quality PDF preferred (either format should be produced directly from original creation package), or original software format)

- 3) Included a 100 word media summary of your paper when requested at submission. Please ensure you have entered correct contact details (email, institution and telephone) in your user account
- 4) Included the raw data to support the claims made in your paper. You can either include your data as electronic supplementary material or upload to a repository and include the relevant doi within your manuscript
- 5) All supplementary materials accompanying an accepted article will be treated as in their final form. Note that the Royal Society will neither edit nor typeset supplementary material and it will be hosted as provided. Please ensure that the supplementary material includes the paper details where possible (authors, article title, journal name).

Best wishes,

Dr Laura Smith
Publishing Editor, Journals

RSC Associate Editor:
Comments to the Author:
(There are no comments.)

RSC Subject Editor:
Comments to the Author:
(There are no comments.)

Reviewer comments to Author:
Reviewer: 1

Comments to the Author(s)
I recommend the publication of this manuscript after the author address the following points. There are several minor defects in this manuscript, please check it carefully and correct them. For example, the repeat spelling of “with” in the first line on page 14.

Author's Response to Decision Letter for (RSOS-191562.R1)

See Appendix C.

Decision letter (RSOS-191562.R2)

10-Dec-2019

Dear Dr Platts:

Title: Quantum Chemical Molecular Dynamics and Metadynamics Simulation of Aluminium Binding to Amyloid- β and Related Peptides
Manuscript ID: RSOS-191562.R2

It is a pleasure to accept your manuscript in its current form for publication in Royal Society Open Science. The chemistry content of Royal Society Open Science is published in collaboration with the Royal Society of Chemistry.

RSC Associate Editor
Comments to the Author:
(There are no comments.)

Reviewer(s)' Comments to Author:

Appendix A

I have now revised the manuscript in the light of referees' comments, with details below, and all significant changes to the MS highlighted in yellow.

Reviewer 1

Specificity is not considered and discussed well

It is not clear to me what this refers to: specificity for different sites within a single peptide, or between different peptides. If the former, then we have provided evidence for preferred binding sites in Table 2. If the latter, then this is an interesting point, but one that work such as this cannot answer. Simulation of multiple Al(III) ions and several peptides *might* be able to address this, but is beyond the scope of this paper and also beyond the computational resources currently available to us.

Buffer environment and pH transition is not included as well

This is true, but could also be a criticism of almost any biomolecular simulation study. Constant pH simulations are possible (though still rare) with conventional MM, but not currently with quantum chemical methods. We are therefore constrained to keeping a fixed protonation state throughout simulations, especially with restraints on bonds to hydrogen. This is a standard protocol in MM, QM and most QM/MM studies. I have added a note on this in the methods section, and potential limitations of this approach were already discussed towards the end of the paper (p 14 of revised version).

Difference between beta-amyloid monomers and oligomers are not covered

This is an interesting point, but beyond the scope of the current study. To keep the calculations tractable I had to truncate the full A β sequence to just the N-terminal 16 residues, so even the full length monomer is beyond current computing resources, such that application to oligomers is not be feasible at present. I tried to make this point in the discussion section in the original version, apparently unsuccessfully: I have tried to beef this up in revision, pp 14-15.

Provide the evidence that the improved computer simulation can better interpret the interaction between beta-amyloid and metal, by solid bench experimental data:

This is a purely simulation study from a single author, whereas this request would involve an entire research project from a team of experimentalists. At present, there is no agreement on how this could be done, nor on suitable theoretical methods that would provide reliable comparison. This study is designed as a proof of principle on such a method, with a view to future evaluation against experiment.

Reviewer 2

More analysis beyond RMSD is needed

I am at a loss on how to respond to this: as well as RMSD, we already included energy, temperature, radius of gyration, radial distribution functions, structures and binding selectivity data. RMSD is a vital measure in all MD studies as a measure of equilibration, and even more so in work such as this as it is the driver for meta-dynamics simulations to escape local energy wells, and so necessarily features heavily in analysis.

No validation of the used models

Validation against DFT is the very first section of results reported: I hope it is not necessary, in 2019, to defend DFT with dispersion correction and triple- ζ basis set as a suitable method for description of metal-peptide binding. As noted above, experimental validation is well beyond the scope of a single simulation study.

No discussion on the limitations of their approach

This is a fair point: some attempt at this was included in the original version, but perhaps did not go far enough. I have therefore added further discussion on this to pp 14-15 of the revision.

Omit many important details

I am not sure which “important details” this reviewer feels are missing. I included those that I thought most relevant in the main paper, more in ESI, and have also uploaded all relevant trajectories to a public repository. Without more detail on the details required, I do not see a way to respond to this.

Provide no justice to the vast literature on this topic:

The literature is indeed vast, if one takes the topic to cover Alzheimer’s, $A\beta$ and metal ions, far too much to include in the introduction of a short research paper. If we restrict this to just Al though, there is rather less, and if to computational studies then there are actually very few. Nevertheless, I have included more references to the introduction.

Most of the figures have a distance axis denoted as “A” which should be Å.

Done

Figure 3 labeling a and b are missing.

Done

The irregular radial distribution function calculated for Al—O distance from solvent (Figure 3b) could be due to insufficient sampling. Replication of each experiment (at least 3 times) is needed to verify such irregular changes.

I agree that the original plot looks rather irregular, however it is important to note the scale of this plot: essentially very few water molecules spend significant time in contact with the metal ion, so the inevitable variations that result from (meta)dynamics simulations appear to be amplified. Given this fact, sufficient sampling to yield smooth RDF would take much longer than our computing resources will allow. However, to check this further, I have run a third replica of this simulation, and included comparisons of RDF from individual runs in ESI, the new combined plot (which is slightly smoother) in main text, along with some discussion of this point.

Figure 4 images taken from vmd snapshot need labeling. Label amino acids and increase the image resolution. Authors compared conformational changes in peptide from their conventional MD and metadynamics results at different time scale. The images shown in figure 4 shows an unstructured peptide in both. Provide a superimpose structure derived from a fixed time scale of simulation.

I have attempted to improve both the images for Figure 4 and the description in the legend and text. However, a figure with structures overlaid as suggested is very difficult to interpret due to the change in coordination. Moreover, I fundamentally disagree with the suggestion to use fixed timescale: metadynamics accelerates the simulation, such that a nanosecond of conventional MD is not equivalent to the same of metadynamics. The point of the image is to demonstrate how only the latter changes the coordination sphere of Al(III); if interested readers wish to know more about the specific structures, all trajectories have been deposited as open data in readily accessible format.

I hope this work is now acceptable for publication in your journal, and look forward to hearing from you in due course.
Yours faithfully, Jamie Platts

Appendix B

School of Chemistry
Cardiff University
28th October 2019

Dear Editor

Many thanks for handling the manuscript Quantum Chemical Molecular Dynamics and Metadynamics Simulation of Aluminium Binding to Amyloid- β and Related Peptides, RSOS-191562. I have adapted this following the most recent set of comments, with details below, and hope that this is now suitable for publication.

Reviewer 1

Discuss why the A β 1-16 was selected for MD, but not the hydrophobic core of A β 17-22?

I have added a brief explanation of the selection of system for study to p3 of the revised version.

Binding free energy analysis and residue contributions of peptides to the interaction with Al(III) can be calculated and discussed

I deliberately did not include binding (free) energy analysis as it is not currently clear how one would go about this with available software tools. We are actively researching suitable protocols for doing so, for Al and other metals, at present and hope to be in a position to report this soon, but at present this is beyond the scope of this study. I have added a comment on this to p14 of the revision.

Yours faithfully

Jamie Platts

Appendix C

School of Chemistry
Cardiff University
3rd December 2019

Dear Editor

In the light of the latest comments on manuscript Quantum Chemical Molecular Dynamics and Metadynamics Simulation of Aluminium Binding to Amyloid- β and Related Peptides, RSOS-191562, I have carefully read the MS and fixed the single error noted and altered text in one or two places where I thought phrasing could be improved. All changes have been tracked.

I hope that this is now suitable for publication. Yours faithfully

Jamie Platts